# Impact of imperceptible motion delay in avatar head movement away from a target on preference formation

Zixiang Yue[1]*, Yoshihiro Nakata[2]*, Hiroshi Ishiguro[1]

**1** Department of Systems Innovation, Graduate School of Engineering Science, Osaka University, Toyonaka, Osaka, Japan, **2** Department of Mechanical and Intelligent Systems Engineering, Graduate School of Informatics and Engineering, The University of Electro-Communications, Chofu, Tokyo, Japan

* yue.zixiang@irl.sys.es.osaka-u.ac.jp (ZY); ynakata@uec.ac.jp (YN)

**Data availability statement:** The data of this study is available from Figshare. It is available at

## Abstract

This study investigated the effect of imperceptible motion delay in avatar head movements on preference formation among operators. Despite the growing importance of avatars in various applications, there is a notable lack of research on how subtle manipulations of avatar movements influence operator cognition and decision-making. By aligning the head and gaze directions, we developed a virtual avatar system designed to subtly manipulate the operator's attention toward specific objects using motion delay. The experimental results revealed that operators tend to prefer objects presented without motion delay, suggesting a negative effect of motion delay on preference formation. These findings highlight motion delay as a significant factor in directing attention and shaping preferences in virtual environments, which have implications for enhancing user experience and optimizing human–computer interaction design. It also emphasizes the necessity of protective measures in avatar-based systems to guard against subtle influences and underscores the ethical and security concerns related to these technologies.

## Introduction

Avatars, which represent operators in specific environments [1], are widely used in remote manipulation scenarios, such as online education [2–4] and virtual social communications [5–7]. They are also earmarked for advanced applications, including remote surgery [8,9] and exploration in hazardous environments [10,11]. Functioning as extensions of the human body [12–14], avatars offer high operability and the ability to overcome physical distances, while ensuring operator safety [10,15]. Consequently, their importance has grown significantly, particularly in scenarios involving teleoperation tasks [16,17].

Studies have shown that human cognition, including decisions [18] and preferences [19], is influenced by external interference. As extensions of the operator's body, avatars can potentially be manipulated by third parties—sometimes without the consent of the operator—thereby affecting their cognition. Moreover, when such interference is subtle or imperceptible, operators may be unknowingly influenced by third parties, while mistakenly believing that their decisions or preferences are entirely self-driven. Despite the significance of this issue

the following URL: https://figshare.com/s/91351a3e40c26e35ae9f.

**Funding:** This work was funded by Japan Science and Technology Agency Moonshot Research and Development Program (grant #JPMJMS2011). The URL is "https://www.jst.go.jp/moonshot/en/index.html". YN and HI were the authors who received award from the mentioned funder. No sponsors or funders had a role in the study design, data collection and analysis, decision to publish, or preparation of the manuscript.

**Competing interests:** The authors have declared that no competing interests exist.

in the context of avatar usage, there is a notable lack of extensive research on the third-party control of avatars, their associated risks, and their impact on operators.

In our previous study, we explored the influence of imperceptibly manipulating the head movements of a teleoperated avatar on the preference formation of operators for human faces [20]. After observing two computer-generated faces multiple times through our virtual teleoperated avatar system, the operators were instructed to choose their preferred face from the two presented options. The head direction of the avatar was synchronized with that of the operator, after which it was subtly guided toward a specific target, making it easier to observe the target. Our findings revealed that operators developed a preference for the target face toward which the avatar was guided and were unaware of our interference. This demonstrates that directing the head movement of an avatar toward a specific target can influence operator preference formation.

However, a limitation of our previous study was the need to determine three parameters: the imperceptible speed of the avatar's head when the operator's head was stationary, moving toward the target, or moving away from the target. The measurement of these parameters is complex, which may hinder the practical application of the proposed method. However, motion delay—an inherent characteristic of teleoperation systems [21] that disrupts operators' intentional movements—is often regarded as a negative factor impacting user experience in avatar operations [22,23], particularly when the delay is significant. Although motion delay has traditionally been viewed as a negative factor in teleoperation systems, it is worth investigating whether such delays—particularly when subtle and imperceptible—can be positively applied. For example, motion delay may be leveraged to influence human cognition, such as shaping perceptual judgments or guiding preference formation during observational tasks.

In this study, we proposed a method that leverages motion delay to prevent the operator from looking away from a target, thereby facilitating the development of a preference for that target. Compared to our previous study—which required a complex measurement of three separate parameters to ensure the imperceptibility of head movement manipulation—the current approach simplifies the process by using only a single parameter: motion delay. Notably, although motion delay is typically considered a negative factor that should be minimized in teleoperation systems, our approach utilizes it constructively. Specifically, this delay negatively affects preference formation toward non-target objects. By introducing a subtle motion delay when the operator attempts to shift their gaze away from the target, we hypothesized that this delay would make it harder to observe the non-target in that direction, thereby enhancing the preference of the operator for the target. We aimed to explore the potential of motion delay as a subtle and effective mechanism for influencing operator behavior. To achieve this, we developed a system capable of introducing motion delays, specifically when the head of the avatar moves away from the target object. In the main experiment, we aimed to intentionally influence preference formation toward computer-generated faces using these imperceptible motion delays. Specific delay values were determined based on the results of a pilot study. Our findings demonstrated that imperceptible motion delays in avatar operations can be used to intentionally influence preference formation.

This study makes two key contributions to the literature. First, it introduces a novel method for subtly influencing preference formation via motion delay in teleoperated avatars. Second, it experimentally validates the effectiveness of the proposed method in influencing the preference formation of operators toward computer-generated faces.

## Materials and methods

### Head-movement intervention method using motion delay

We propose a head-movement intervention method that introduces motion delay to subtly prevent avatar head movement, denoted as $Head_{avtr}$, from aligning with operator head movement, denoted as $Head_{opr}$ (Fig 1). As illustrated in Fig 1a, the operator wears a head-mounted display and observes objects, including the target object, through a virtual avatar system. Our proposed method introduces a motion delay when the operator attempts to look at a non-target object. Therefore, in the context of observing two objects, no motion delay is introduced when $Head_{opr}$ moves toward the target object or is considered stationary. During this process, the movement of $Head_{avtr}$ was fully synchronized with that of $Head_{opr}$, thereby ensuring ease of operator observation. However, when the operator moves the head away from the target to observe another object, an imperceptible motion delay is applied to the movement of $Head_{avtr}$. This delay temporarily halts the movement of $Head_{avtr}$ for a brief period, resulting in a slight desynchronization between $Head_{opr}$ and $Head_{avtr}$ (Fig 1b). Once the delay period ends, $Head_{avtr}$ starts following the trajectory of $Head_{opr}$, allowing the operator to perceive its movement as natural (Fig 1c). This continues until $Head_{avtr}$ and $Head_{opr}$ are fully synchronized (Fig 1d). As a result, the operator will find it more challenging to observe the nontarget object owing to the motion delay. The algorithm used in our proposed method, which introduces an imperceptible motion delay to intervene with the avatar's head movement, is detailed in S1 Appendix.

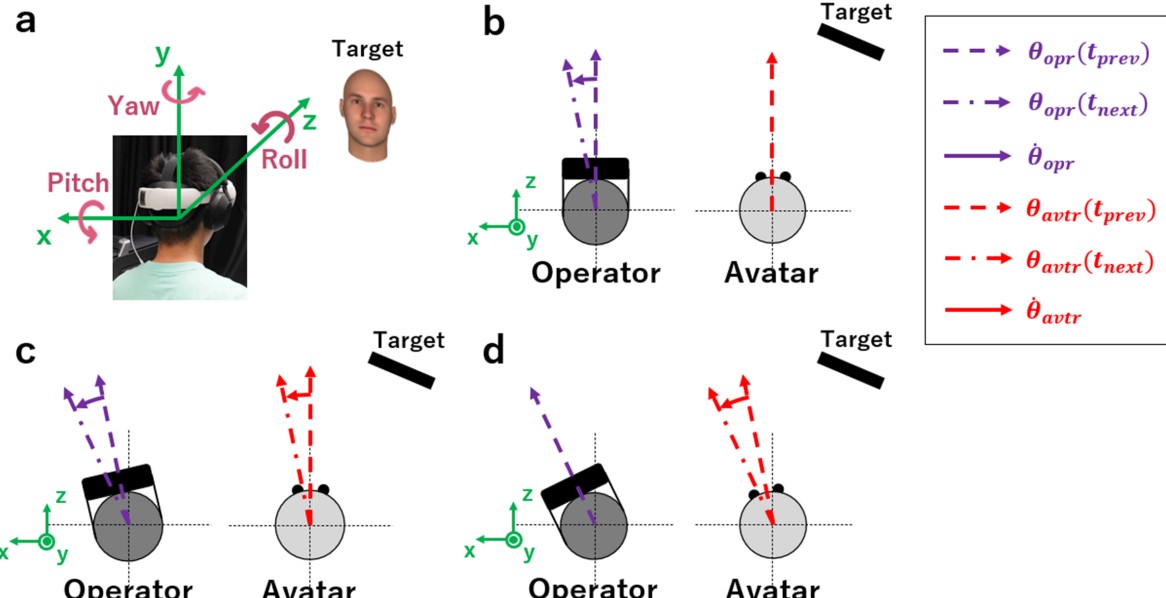

**Fig 1. Head-movement intervention method using motion delay.** For simplicity, our method applies motion delay only to the yaw rotation of avatar head. $\theta_{opr}(t)$ and $\theta_{avtr}(t)$ denote the yaw rotation angles of the head of the operator and avatar at a particular time point $t$, respectively. Similarly, the angular velocities of the yaw rotation at time point $t$ are represented as $\dot{\theta}_{opr}(t)$ and $\dot{\theta}_{avtr}(t)$, corresponding to the operator and avatar, respectively. $t_{prev}$ and $t_{next}$ are two consecutive time points, respectively. (a) The operator wears a head-mounted display and observes objects, including the target object, through a virtual avatar system. The pitch, yaw, and roll rotations of the head are shown. (b) When the operator rotates their head away from the target, an imperceptible motion delay is introduced. The head of the avatar remains stationary during the motion-delay period. (c) Once the motion-delay period ends, the head of the avatar replicates the trajectory of the head movements of the operator. (d) This process concludes when the head of the avatar is synchronized with that of the operator.

In our virtual avatar system, $Head_{opr}$ is considered stationary if the angular velocity of the head is less than or equal to the threshold value $Thr\_vel$. This was designed to prevent minute head movements from triggering motion delays, which could lead to difficulties in head movements. This threshold was set to 5 degrees per second, as referenced from our previous study [20]. Furthermore, our proposed method relies on two prerequisites, which are outlined below.

1. Our proposed method is specifically designed for compatibility with our virtual avatar system, which necessitates that the line of sight of the operator remains aligned with their head movement. Consequently, during the experiment, operators were required to maintain their gaze fixed at the center of their field of view, and only observe objects by rotating their heads.
2. For simplicity, we limited the head-movement intervention via motion delay to yaw rotation—the horizontal plane rotation—within our virtual avatar system (Fig 1a). This approach is considered natural when the motion delay is minimal.

## Virtual environment

We developed a virtual avatar system for observing facial images by incorporating the proposed head-movement intervention method using motion delay. The system was built using Unity 5.6.7f1 and operated on a computer (G-Tune LITTLEGEAR i3, MouseComputer CO., LTD). In this system, $Head_{avtr}$ was represented by the main camera, which was positioned at the center of a cylindrical object with a radius of 4.6 units and a height of 70.71 units in Unity. The pitch, yaw, and roll angles of the camera were limited to $\pm50°$. A forest image was displayed on the inner wall of the cylinder as the background, alongside two computer-generated faces. Initially, the faces were positioned such that the center of each face was 22.5° from the main camera, creating an angle of 45° between them. At 4.6 units, the distance from the main camera to each face matched the cylinder radius. Because instantly controlling the movement of the main camera is inconvenient, the camera was kept stationary, while the cylindrical object, including the face images and background, was rotated to simulate the movement of $Head_{avtr}$. Furthermore, our head-movement intervention method blocked the yaw rotation of the cylindrical object in a given period to simulate the motion delay in the movement of $Head_{avtr}$.

## Computer-generated faces

Following the methodology outlined by Shimojo et al. [19], we generated 40 male and 40 female European facial images using FaceGen. Starting with the generic European faces, we sequentially adjusted the parameters "Brow Ridge High/Low," "Eyes Down/Up," "Mouth Lips Large/Small," and "Nose Down/Up." Each parameter was assigned a random value ranging from -0.5 to 0.5 (S2 Appendix).

In this study, European faces were selected instead of East Asian faces. This decision was based on the fact that all participants were Japanese students who may have had pre-existing preferences or biases toward East Asian faces. We anticipated that such pre-existing preferences could interfere with the effectiveness of our proposed method in influencing preference formation. Therefore, we generated European faces, which are generally considered more difficult for Japanese participants to distinguish from one another.

## Survey form

Before conducting the preference formation influence experiment, the participants were asked to rate the attractiveness of 40 male face images, followed by 40 female face images. The survey was created using Google Forms, with each page displaying a single face image. The participants were instructed to rate the attractiveness of each face on a scale of 1 to 7 (S3 Appendix).

## Experimental setup

A head-mounted display (HMD; Oculus Quest 2, Oculus VR, Inc.) was used to observe the stimuli in a virtual environment. The operators were required to press one of the two buttons to respond to the stimuli. Additionally, a pair of headphones (SONY WH-1000XM5, Sony Group Corporation) was used to play white noise during the experiments. A photograph of the experimental setup is presented in Fig 2a, and 2b shows a schematic of the setup.

## Participants

The participants were 26 students (12 males and 14 females) from Osaka University and other universities in Osaka, Japan. The number of participants was set to match that in our previous study [20], as both studies investigated a similar issue using different methods. The participants were recruited via the Kantan Open Access Network of Osaka University and the X (formerly Twitter) account of the Ishiguro Laboratory between February 16, 2024, and March

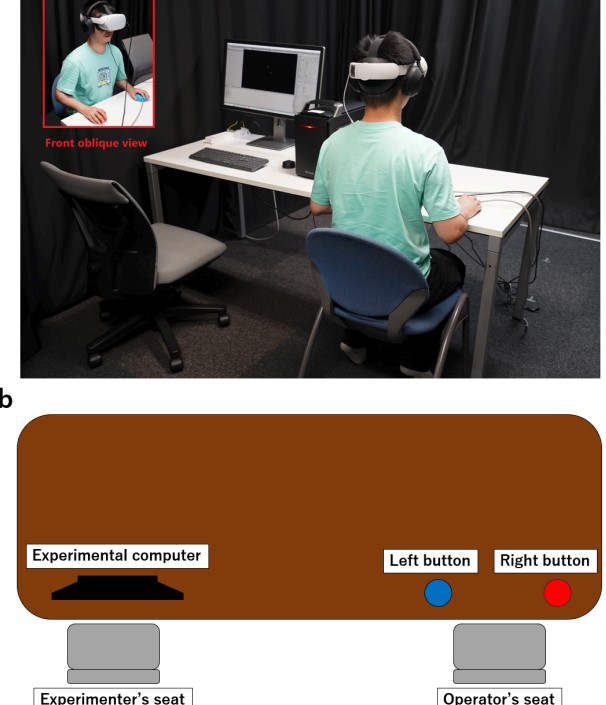

**Fig 2. Experimental setup.** (a) Photograph of experimental setup. The individual shown in the photograph is not an actual participant in the study but a volunteer who assisted in demonstrating the setup. (b) Schematic of the experimental setup.

15, 2024. They were aged between 18 and 26 years (*mean* = 21.19, *standard deviation* = 1.98). All participants were given 1,300 yen to participate and were assigned to perform the same tasks. Participants who participated in this experiment were required to have a corrected visual acuity greater than 0.7. We excluded two responses: one because a male participant sensed the motion delay during the experiment, and another because a female participant did not follow the instructions to observe each object at least five times. For the final data analysis, 24 participants performed the experimental tasks.

## Preference formation influence experiment

In this study, we investigated whether preference formation toward computer-generated faces can be influenced by introducing an imperceptible motion delay in avatar head movements. Our hypothesis is that when participants compare two facial images and select their preferred one, they are more likely to select the face without motion delay, even if they are unable to consciously perceive this delay.

**Rating phase.** Each participant was instructed to rate the attractiveness of the facial images in the survey. Ten male and ten female face images with ratings closest to the median for each gender were selected. These selected images were paired within their respective genders based on the most similar ratings, resulting in ten pairs of facial images for use in the experiment.

**Instruction phase.** Before conducting the preference formation influence experiment, participants were instructed on key points to adhere to during the facial image observations. First, they were asked to keep their body oriented forward, allowing only their head to rotate to ensure that their gaze remained centered. Second, they were instructed to rotate their heads to position each face image at the center of their field of view while maintaining a fixed gaze. Finally, the participants were required to alternately observe the two faces for at least five repetitions, ensuring that the head movements between the images were neither negative nor too slow.

**Experimental phase.** After the Instruction Phase, the operator wore an HMD and headphones. White noise was played through headphones during the entire Experimental Phase. In each trial, the operators began by positioning a white cross at the center of their field of view using head movements. Thereafter, a pair of face images selected from the Rating Phase was presented on the left and right sides. Throughout the Experimental Phase, when the operator attempted to rotate their head to observe the non-target face image, the yaw rotation of the avatar head was momentarily halted for 80 ms. This duration, determined in the preliminary experiments, was considered imperceptible within our virtual avatar system. The target face image did not include a motion delay. Additionally, the target face image was earmarked for selection as the preferred face. Once the operator decided which face they found more attractive, they pressed the corresponding button (e.g., pressing the left button if they preferred the face on the left side) to complete the trial. Ten trials were conducted, with the direction of the target faces randomized across the trials. After completing the experiment, the participants were asked if they felt any sense of unnaturalness during the experiment and to explain their reasons for their preferred face choices.

## Data collection and processing

In the preference formation influence experiment, operators were required to rotate their head to observe each face image at least five times per trial. For each trial, we recorded the yaw rotation angles of both $Head_{opr}$ and $Head_{avtr}$ at each time step. Additionally, to evaluate the extent to which the operator's choices aligned with the target faces, we recorded the direction of the target face and the operator's final selection for each trial.

For each trial, we plotted the yaw rotation angles of both $Head_{opr}$ and $Head_{avtr}$ over time to visualize their movement paths. To ensure data quality, we excluded an operator's data if they failed to observe each face image at least five times in three or more trials. This criterion was used to ensure sufficient engagement with stimuli. To evaluate how well the operators' final selections matched the target faces, we counted the number of trials (out of ten) in which each operator selected the target face. These counts were then compared to the expected frequency under a random selection scenario (50%) using a one-tailed paired-samples t-test to evaluate the impact of the intervention. This analysis was conducted in Python using the function `scipy.stats.ttest_rel`. Prior to the t-test, the normality of the data distribution was assessed using the Shapiro–Wilk test, implemented via `scipy.stats.shapiro`. In addition to assessing statistical significance, we calculated Cohen's d to evaluate the size of the effect (S4 Appendix).

## Ethics

The Ethics Committee for Research Involving Human Subjects at the Graduate School of Engineering Science, Osaka University approved the study protocol (#R2-10-1). All participants provided written informed consent before participating in the experiment. The experiments were performed in accordance with the principles and guidelines of the Declaration of Helsinki (1975).

Informed consent was obtained from the volunteer depicted in Fig 2a for the publication of the image in this open-access paper. The individual in the photograph was not an actual participant, but volunteered to demonstrate the experimental setup.

## Results

The results of the preference formation experiment are shown in Fig 3. On average, participants selected the target face as their preferred option 59.2% of the time, exceeding the 50% chance level. The Shapiro–Wilk test confirmed that the data did not significantly deviate from normality, $p = 0.393$. A one-tailed paired-samples t-test showed that participants selected the target face significantly more often than by chance ($M = 59.2\%$, $SD = 1.56$), $t(23) = 2.88$, $p = 0.004$, Cohen's d = 0.83, indicating a large size effect.

## Discussion

By leveraging the inherent motion delay in teleoperation systems, we proposed a head-movement intervention method to control the head movements of the avatar (Fig 1). The operators were asked to observe two computer-generated faces and choose their preferred face. The proposed method introduced an imperceptible motion delay when operators attempted to rotate their heads away from the target face—the face image intended for selection as the preferred choice. The experimental results demonstrated that operators were more likely to choose the target face—which was unaffected by motion delay—as their preferred choice after multiple comparisons. This suggests that imperceptible motion delays can effectively influence preference formation for computer-generated faces (Fig 3). A possible explanation is that the motion delay made it more challenging to observe the non-target face, even though the delay was imperceptible. Therefore, we suggest that motion delays shape preference formation by making it more difficult for operators to look away from the target face, thereby indirectly influencing their choices.

The proposed method has some limitations. For simplicity, we introduced a motion delay only for the yaw rotation of $Head_{avtr}$. In our preference formation experiment, only one of the

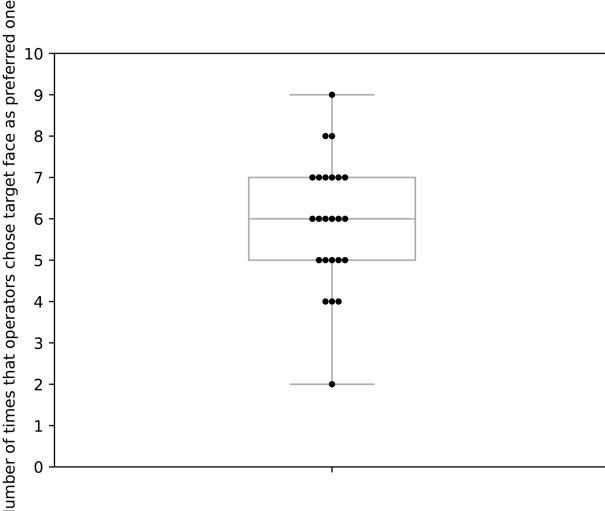

**Fig 3. Frequency of operators choosing the target face as the preferred one.**

26 participants reported feeling unnatural, suggesting that this approach is generally acceptable. However, a more natural implementation would involve applying a motion delay to all head rotations, including the pitch, yaw, and roll. Additionally, instead of measuring the precise threshold of motion delay that is both effective in influencing preference formation and remains imperceptible, we set the motion delay to 80 ms based on the results of the preliminary experiments. Furthermore, to avoid scenarios in which subtle head movements could trigger motion delays—causing operators to frequently experience difficulty in head movement during observation—we configured the system to ensure that no motion delay occurred when the angular velocity of $Head_{opr}$ was less than or equal to 5 degrees per second. Consequently, our system could not influence the preference formation of the operators when the movements of $Head_{opr}$ were relatively slow during the observation process. To address this problem, we instructed the participants to avoid excessively slow head movements during the primary experiment. However, we believe that this issue can be resolved by refining the motion-delay mechanism.

The preference formation influence experiment had several limitations. First, anticipating that Japanese participants' pre-existing preferences for East Asian faces might reduce the effectiveness of our proposed method, we specifically generated European faces, which are generally considered more difficult for Japanese participants to distinguish than East Asian faces. Additionally, these facial images were intentionally designed to be highly similar in appearance to align with the same objectives. Second, because our system cannot control the gaze of the avatar, operators were instructed to fix their gaze at the center of their field of view and rotate their heads until a face was at the center of their observation. Third, operators were asked to observe each pair of face images at least five times before selecting their preferred choice to ensure that they experienced motion delay multiple times before making their decision.

In our previous study [20], we imperceptibly guided the head of the avatar toward the target face to facilitate observation, thereby guiding the operators to choose the target face as the preferred face. However, this method requires the complex measurement of three parameters to ensure that the manipulation remains imperceptible, which makes practical application

difficult. Therefore, reducing the number of required parameters is beneficial. In the present study, we aimed to achieve a similar effect on preference formation using a simpler approach that relies on only one parameter: motion delay. Additionally, although motion delay is typically regarded as a negative factor in teleoperation systems—something to be minimized—we intentionally utilized it to influence preference formation toward the target objects. Our findings demonstrate the potential for motion delay. To determine which of the two approaches had a stronger effect on preference formation, we conducted an independent samples t-test to compare the means of the two conditions. There was no significant difference between the groups, $t(50) = -0.553$, $p = 0.583$. Therefore, statistical analyses indicated no significant difference between the proposed methods in these two studies, suggesting that their effects on preference formation were comparable. Considered together with our previous research, this study highlights the possibility of influencing operator cognition indirectly through the manipulation of avatar movements, rather than through direct intervention with the operators themselves.

Motion delay is an inherent issue in teleoperational systems. Our research highlights the influence of imperceptible motion delays on preference formation toward computer-generated operator faces when using teleoperated avatars in a virtual environment. One avenue for future work is to test whether this mechanism could also influence the preference formation of operators when observed using a real teleoperated robot. In addition, we can investigate whether other consciousness-related behaviors are also affected by our proposed method. Furthermore, online training systems can leverage motion delays to enhance the learning process by subtly introducing delays when users perform undesired actions, thereby guiding them toward more effective behaviors. Similarly, applications for social interaction support may employ motion delay to indirectly increase the salience or attractiveness of specific objects or individuals. Moreover, as suggested by our results, developers of such systems should not only aim to minimize motion delays as much as possible but also enhance system security to prevent third parties from exploiting these vulnerabilities.

## Conclusion

In conclusion, our study revealed the potential for imperceptibly manipulating operator preferences through subtle motion delays in avatar operations. Avatars are frequently regarded as outward extensions of an operator's body. However, this connection allows a third party to influence the operator's preferences. By introducing motion delay, a feature commonly present in teleoperation systems, the preference formation toward computer-generated faces was subtly influenced by making non-target faces more difficult to observe.

Therefore, our findings underscore the negative impact of imperceptible motion delay on preference formation during avatar operation. Apart from shaping preferences, this approach could also influence other cognitive functions, highlighting the broader implications for human–computer interaction and ethical considerations surrounding such technologies. Potential applications include online training systems that introduce motion delays when the operator performs undesirable actions and social interaction support tools that subtly enhance the visual attraction of specific objects. Thus, future studies could explore the impact of imperceptible motion delay on various cognitive functions beyond preference formation, investigate the broader implications for human–computer interaction, address the ethical considerations surrounding the use of such technologies, and focus on developing applications that leverage motion delay in avatar-based systems.

## Supporting information

**S1 Appendix. Algorithm: Head-movement intervention using motion delay.** Pseudocode describing the proposed method of head-movement intervention using motion delay.
(DOCX)

**S2 Appendix. Parameters for generating computer-generated faces.** We generated 80 computer-generated faces by sequentially adjusting the parameters "Brow Ridge High/Low," "Eyes Down/Up," "Mouth Lips Large/Small," and "Nose Down/Up." The actual values of the parameters were recorded in a CSV file.
(CSV)

**S3 Appendix. Survey forms for rating the attractiveness of computer-generated faces.** The description and an example of the survey forms for rating the attractiveness of computer-generated faces are listed in this file.
(DOCX)

**S4 Appendix. Python code for calculating Cohen's d.** Python implementation for calculating Cohen's d, which is a measure of effect size.
(DOCX)

## Acknowledgments

We thank the two reviewers for their excellent feedback, which has improved this manuscript.

## Author contributions

**Conceptualization:** Zixiang Yue, Yoshihiro Nakata, Hiroshi Ishiguro.

**Data curation:** Zixiang Yue.

**Formal analysis:** Zixiang Yue.

**Funding acquisition:** Yoshihiro Nakata, Hiroshi Ishiguro.

**Investigation:** Zixiang Yue.

**Methodology:** Zixiang Yue, Yoshihiro Nakata.

**Project administration:** Yoshihiro Nakata, Hiroshi Ishiguro.

**Software:** Zixiang Yue.

**Supervision:** Yoshihiro Nakata.

**Validation:** Zixiang Yue.

**Visualization:** Zixiang Yue, Yoshihiro Nakata.

**Writing – original draft:** Zixiang Yue, Yoshihiro Nakata.

**Writing – review & editing:** Zixiang Yue, Yoshihiro Nakata, Hiroshi Ishiguro.

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
