## [Decision Letter · Decision Letter 0]

8 May 2025

PONE-D-25-07431Impact of imperceptible motion delay in avatar head movement away from a target on preference formationPLOS ONE

Dear Dr. Yue,

Thank you for submitting your manuscript to PLOS ONE. After careful consideration, we feel that it has merit but does not fully meet PLOS ONE’s publication criteria as it currently stands. Therefore, we invite you to submit a revised version of the manuscript that addresses the points raised during the review process.

Please provide detailed and point-by-point response to all the comments.

We look forward to receiving your revised manuscript.

Kind regards,

Dr. Mohammad Mofatteh, PhD, MPH, MSc, PGCert, BSc (Hons), MB BCh (c)

Academic Editor

PLOS ONE

[This work was supported by the Japan Science and Technology Agency Moonshot Research and Development Program under Grant JPMJMS2011.]

 [This work was funded by Japan Science and Technology Agency Moonshot Research and Development Program (grant #JPMJMS2011). The URL is "https://www.jst.go.jp/moonshot/en/index.html". YN and HI were the authors who received award from the mentioned funder. No sponsors or funders had a role in the study design, data collection and analysis, decision to publish, or preparation of the manuscript.]

4. We are unable to open your Figure file [Fig3.eps]. Please kindly revise as necessary and re-upload.

Additional Editor Comments:

Please provide detailed and point-by-point response to all the comments.

Reviewers' comments:

Reviewer's Responses to Questions

**Comments to the Author**

1. Is the manuscript technically sound, and do the data support the conclusions?

Reviewer #1: Yes

Reviewer #2: Yes

2. Has the statistical analysis been performed appropriately and rigorously? 

Reviewer #1: Yes

Reviewer #2: Yes

3. Have the authors made all data underlying the findings in their manuscript fully available?

Reviewer #1: Yes

Reviewer #2: Yes

4. Is the manuscript presented in an intelligible fashion and written in standard English?

Reviewer #1: Yes

Reviewer #2: Yes

5. Review Comments to the Author

Reviewer #1: 1. The introduction gives a general overview but does not explain the problem addressed or why the proposed method is novel in comparison to earlier work.

2. The article does not provide comprehensive descriptions of the steps involved in data collection and signal processing; this should be improved.

3. The statistical methods used for data analysis are not explained in detail; it should be explained more thoroughly how the tests were conducted and whether the expectations of the proposed method were met.

4. The results section clearly presents the findings, but a more thorough discussion of statistical significance and effect sizes is advised.

5. The main text does not adequately explain certain figures. It is advised that the narrative include explicit citations and interpretations for all figures and tables.

6: The discussion section does not thoroughly examine the ramifications of the findings and is somewhat short. It is necessary to conduct a more thorough examination of how these findings add to the corpus of current knowledge.

Reviewer #2: The paper is original scientific in nature and concerns the delay of avatar head movement away from the target for preference formation.

For a better clarification, please edit your paper as follows:

However, it should be noted that the authors should make some corrections in the article.

1. Fig. 1 should be expanded for references that should be included in the content - all other drawings have one, and the specificity of the presented system even requires that should be presented in this way.

The readability of the course shown in Figure 1 should be improved. In addition, the results shown in Figure 1 have been described too scarcely, it is recommended to extend the description of point Head-movement intervention method using motion delay (page 2), so that the reader can obtain more information.

2. Expand the text of the manuscript (or the introduction or conclusion) with specific results in the world and in Europe, - increase the quality of the work by listing the results of publications of researchers and experts working in this field registered in world databases. These are: Analysis of the possibilities of tire-defect inspection based on unsupervised learning and deep learning (21, DOI 10.3390/s21217073), Collaborative assembly task realization using selected type of a human-robot interaction (DOI 10.1016/j.trpro.2019.07.078), Implementation of predictive models in industrial machines with proposed automatic adaptation algorithm (DOI 10.3390/app12041853) and Solutions to the characteristic equation for industrial robot's elliptic trajectories (DOI10.17559/TV-20150114112458) thanks.

The presented literature is accurate, but most of the indicated items are from one country (Japan), it is recommended to extend it by several additional foreign items.

3. figures 1should be contrasting and readable,

4. conclusions and future work should be extended to contain practical applications based on research described in this paper - expand references,

5. if possible, highlight the course of dependencies/relations in figure No. 1 - the yellow color is indistinct,

6. For ARTICLE type, 19 references are not enough. Please add more references (>19) during your revisions.

I recommend publishing the post after the proposed modifications.

6. PLOS authors have the option to publish the peer review history of their article (what does this mean?). If published, this will include your full peer review and any attached files.

Reviewer #1: No

Reviewer #2: **Yes: **Pavol Bozek

---

## [Author Response · Author response to Decision Letter 1]

1 Jul 2025

Response to Editor:

Comment 1

Answer

Thank you for your suggestion regarding the deposition of laboratory protocols on protocols.io. We appreciate the opportunity to improve the reproducibility of our study. However, we believe that the experimental procedures and system architecture have already been described in sufficient detail in the main text of our manuscript. Therefore, we would prefer to proceed without submitting a separate protocol to protocols.io at this time. Please let us know if further clarification is required.

Comment 2

Answer

Based on this requirement, we downloaded the LaTeX template from https://journals.plos.org/plosone/s/latex. Our manuscript has been formatted accordingly using this template, and the naming conventions for the figures and appendices have also been adjusted to comply with the submission system's requirements.

Comment 3

[This work was supported by the Japan Science and Technology Agency Moonshot Research and Development Program under Grant JPMJMS2011.]

[This work was funded by Japan Science and Technology Agency Moonshot Research and Development Program (grant #JPMJMS2011). The URL is "https://www.jst.go.jp/moonshot/en/index.html". YN and HI were the authors who received award from the mentioned funder. No sponsors or funders had a role in the study design, data collection and analysis, decision to publish, or preparation of the manuscript.]

Answer

Based on this suggestion, we removed “This work was supported by the Japan Science and Technology Agency Moonshot Research and Development Program under Grant JPMJMS2011” from the Acknowledgments section and revised it as “We thank the two reviewers for their excellent feedback, which has improved this manuscript.”

Comment 4

Answer

Based on this suggestion, we uploaded the dataset to Figshare. It is available at the following URL: https://figshare.com/s/91351a3e40c26e35ae9f.

Comment 5

4. We are unable to open your Figure file [Fig3.eps]. Please kindly revise as necessary and re-upload.

Answer

We appreciate your bringing this issue to our attention. We have recreated Fig3.eps and confirmed that it can now be opened without any issues. The corrected version of Fig3.eps has been re-uploaded.

Comment 6

Answer

Thank you for bringing this issue to our attention. We have reviewed all the references cited in our manuscript and confirmed that none have been retracted.

Response to Reviewer: 1

Comment 1

1. The introduction gives a general overview but does not explain the problem addressed or why the proposed method is novel in comparison to earlier work.

Answer　

Based on this suggestion, we compared the present study with our previous work to highlight the novelty of the proposed method. Two key reasons are outlined in the introduction. First, our previous approach required the calibration of three parameters to ensure that the manipulation remained imperceptible to the operators. In contrast, the current method requires only a single parameter, which significantly simplifies the implementation of the avatar system. Second, unlike traditional approaches that aim to minimize motion delay because of its negative impact on user experience, our method intentionally leverages subtle motion delay as a tool to achieve cognitive influence.

Introduction, Paragraph 4 is revised as follows:

　　“However, a limitation of our previous study was the need to determine three parameters: the imperceptible speed of the avatar's head when the operator’s head was stationary, moving toward the target, or moving away from the target. The measurement of these parameters is complex, which may hinder the practical application of the proposed method. However, motion delay---an inherent characteristic of teleoperation systems\cite{farajiparvar2020brief} that disrupts operators' intentional movements---is often regarded as a negative factor impacting user experience in avatar operations\cite{claypool2006latency,kim2021impact}, particularly when the delay is significant. Although motion delay has traditionally been viewed as a negative factor in teleoperation systems, it is worth investigating whether such delays---particularly when subtle and imperceptible---can be positively applied. For example, motion delay may be leveraged to influence human cognition, such as shaping perceptual judgments or guiding preference formation during observational tasks.”

Introduction, Paragraph 5 is revised as follows:

　　“In this study, we proposed a method that leverages motion delay to prevent the operator from looking away from a target, thereby facilitating the development of a preference for that target. Compared to our previous study---which required a complex measurement of three separate parameters to ensure the imperceptibility of head movement manipulation---the current approach simplifies the process by using only a single parameter: motion delay. Notably, although motion delay is typically considered a negative factor that should be minimized in teleoperation systems, our approach utilizes it constructively. Specifically, this delay negatively affects preference formation toward non-target objects. By introducing a subtle motion delay when the operator attempts to shift their gaze away from the target, we hypothesized that this delay would make it harder to observe the non-target in that direction, thereby enhancing the preference of the operator for the target. We aimed to explore the potential of motion delay as a subtle and effective mechanism for influencing operator behavior. To achieve this, we developed a system capable of introducing motion delays, specifically when the head of the avatar moves away from the target object. In the main experiment, we aimed to intentionally influence preference formation toward computer-generated faces using these imperceptible motion delays. Specific delay values were determined based on the results of a pilot study. Our findings demonstrated that imperceptible motion delays in avatar operations can be used to intentionally influence preference formation.”

Comment 2

2. The article does not provide comprehensive descriptions of the steps involved in data collection and signal processing; this should be improved.

Answer

Based on this suggestion, we added a subsection “Data collection and processing” in Section “Materials and methods”. We described the collection of data in the first paragraph and subsequently mentioned how to process the data we collected in the second paragraph.

Materials and methods, Data collection and processing, Paragraph 1 is written as follows:

　　“In the preference formation influence experiment, operators were required to rotate their head to observe each face image at least five times per trial. For each trial, we recorded the yaw rotation angles of both $Head_{opr}$ and $Head_{avtr}$ at each time step. Additionally, to evaluate the extent to which the operator’s choices aligned with the target faces, we recorded the direction of the target face and the operator’s final selection for each trial.”

Materials and methods, Data collection and processing, Paragraph 2 is written as follows:

　　“For each trial, we plotted the yaw rotation angles of both $Head_{opr}$ and $Head_{avtr}$ over time to visualize their movement paths. To ensure data quality, we excluded an operator’s data if they failed to observe each face image at least five times in three or more trials. This criterion was used to ensure sufficient engagement with stimuli. To evaluate how well the operators’ final selections matched the target faces, we counted the number of trials (out of ten) in which each operator selected the target face. These counts were then compared to the expected frequency under a random selection scenario (50\%) using a one-tailed paired-samples t-test to evaluate the impact of the intervention. This analysis was conducted in Python using the function \texttt{scipy.stats.ttest\_rel}. Prior to the t-test, the normality of the data distribution was assessed using the Shapiro--Wilk test, implemented via \texttt{scipy.stats.shapiro}. In addition to assessing statistical significance, we calculated Cohen’s d to evaluate the size of the effect (\nameref{S4_Appendix}).”

Comment 3

3. The statistical methods used for data analysis are not explained in detail; it should be explained more thoroughly how the tests were conducted and whether the expectations of the proposed method were met.

Answer

Based on this suggestion, we provided a detailed explanation of how the statistical tests were conducted in the second paragraph of the "Data Collection and Processing" subsection. A one-tailed paired-samples t-test and Shapiro–Wilk test were performed using the Python functions. Cohen’s d was also calculated with Python, and the corresponding code is provided in Appendix S4.

Materials and methods, Data collection and processing, Paragraph 1 is written as follows:

　　“For each trial, we plotted the yaw rotation angles of both $Head_{opr}$ and $Head_{avtr}$ over time to visualize their movement paths. To ensure data quality, we excluded an operator’s data if they failed to observe each face image at least five times in three or more trials. This criterion was used to ensure sufficient engagement with stimuli. To evaluate how well the operators’ final selections matched the target faces, we counted the number of trials (out of ten) in which each operator selected the target face. These counts were then compared to the expected frequency under a random selection scenario (50\%) using a one-tailed paired-samples t-test to evaluate the impact of the intervention. This analysis was conducted in Python using the function \texttt{scipy.stats.ttest\_rel}. Prior to the t-test, the normality of the data distribution was assessed using the Shapiro--Wilk test, implemented via \texttt{scipy.stats.shapiro}. In addition to assessing statistical significance, we calculated Cohen’s d to evaluate the size of the effect (\nameref{S4_Appendix}).”

We explained whether the expectations of the proposed method were met in the “Results” section.

“Results” section is revised as follows:

　　“The results of the preference formation experiment are shown in Fig.~\ref{fig3}. On average, participants selected the target face as their preferred option 59.2\% of the time, exceeding the 50\% chance level. The Shapiro--Wilk test confirmed that the data did not significantly deviate from normality, \emph{p} = 0.393. A one-tailed paired-samples t-test showed that participants selected the target face significantly more often than by chance (\emph{M} = 59.2\%, \emph{SD} = 1.56), \emph{t}(23) = 2.88, \emph{p} = 0.004, Cohen's d = 0.83, indicating a large size effect.”

Comment 4

4. The results section clearly presents the findings, but a more thorough discussion of statistical significance and effect sizes is advised.

Answer

Information on statistical significance and effect sizes is included in the “Results” section.

　　“The results of the preference formation experiment are shown in Fig.~\ref{fig3}. On average, participants selected the target face as their preferred option 59.2\% of the time, exceeding the 50\% chance level. The Shapiro--Wilk test confirmed that the data did not significantly deviate from normality, \emph{p} = 0.393. A one-tailed paired-samples t-test showed that participants selected the target face significantly more often than by chance (\emph{M} = 59.2\%, \emph{SD} = 1.56), \emph{t}(23) = 2.88, \emph{p} = 0.004, Cohen's d = 0.83, indicating a large size effect.”

Comment 5

5. The main text does not adequately explain certain figures. It is advised that the narrative include explicit citations and interpretations for all figures and tables.

Answer

Based on this suggestion, we have included explicit citations and interpretations for all three figures.

Figure 1, mentioned in the first paragraph of Section “Materials and methods”, subsection “Head-movement intervention method using motion delay.”

　　“We propose a head-movement intervention method that introduces motion delay to subtly prevent avatar head movement, denoted as $Head_{avtr}$, from aligning with operator head movement, denoted as $Head_{opr}$ (Fig~\ref{fig1}). As illustrated in Fig~\ref{fig1}a, the operator wears a head-mounted display and observes objects, including the target object, through a virtual avatar system. Our proposed method introduces a motion delay when the operator attempts to look at a non-target object. Therefore, in the context of observing two objects, no motion delay is introduced when $Head_{opr}$ moves toward the target object or is considered stationary. During this process, the movement of $Head_{avtr}$ was fully synchronized with that of $Head_{opr}$, thereby ensuring ease of operator observation. However, when the operator moves the head away from the target to observe another object, an imperceptible motion delay is applied to the movement of $Head_{avtr}$. This delay temporarily halts the movement of $Head_{avtr}$ for a brief period, resulting in a slight desynchronization between $Head_{opr}$ and $Head_{avtr}$ (Fig~\ref{fig1}b). Once the delay period ends, $Head_{avtr}$ starts following the trajectory of $Head_{o

---

## [Decision Letter · Decision Letter 1]

10 Jul 2025

Impact of imperceptible motion delay in avatar head movement away from a target on preference formation

PONE-D-25-07431R1

Dear Dr. Yue,

We’re pleased to inform you that your manuscript has been judged scientifically suitable for publication and will be formally accepted for publication once it meets all outstanding technical requirements.

Kind regards,

Mohammad Mofatteh, PhD, MPH, MSc, PGCert, BSc (Hons), MB BCh (c)

Academic Editor

PLOS ONE

Additional Editor Comments (optional):

Reviewers' comments:

Reviewer's Responses to Questions

**Comments to the Author**

1. If the authors have adequately addressed your comments raised in a previous round of review and you feel that this manuscript is now acceptable for publication, you may indicate that here to bypass the “Comments to the Author” section, enter your conflict of interest statement in the “Confidential to Editor” section, and submit your "Accept" recommendation.

Reviewer #1: All comments have been addressed

Reviewer #2: All comments have been addressed

2. Is the manuscript technically sound, and do the data support the conclusions?

Reviewer #1: Yes

Reviewer #2: Yes

3. Has the statistical analysis been performed appropriately and rigorously? 

Reviewer #1: Yes

Reviewer #2: Yes

4. Have the authors made all data underlying the findings in their manuscript fully available?

Reviewer #1: Yes

Reviewer #2: Yes

5. Is the manuscript presented in an intelligible fashion and written in standard English?

Reviewer #1: Yes

Reviewer #2: Yes

6. Review Comments to the Author

Reviewer #1: The authors have adequately addressed the previous comments and the overall structure and scientific content are clear and coherent.

Reviewer #2: The authors accepted the comments, I recommend the paper to be published.

The authors accepted the comments, I recommend the paper to be published.

The authors accepted the comments, I recommend the paper to be published.

The authors accepted the comments, I recommend the paper to be published.

7. PLOS authors have the option to publish the peer review history of their article (what does this mean?). If published, this will include your full peer review and any attached files.

Reviewer #1: No

Reviewer #2: No

---

## [Editor Report · Acceptance letter]

PONE-D-25-07431R1

PLOS ONE

Dear Dr. Yue,

I'm pleased to inform you that your manuscript has been deemed suitable for publication in PLOS ONE. Congratulations! Your manuscript is now being handed over to our production team.

Kind regards,

on behalf of

Dr. Mohammad Mofatteh

Academic Editor

PLOS ONE